# Action observation intervention using three-dimensional movies improves the usability of hands with distal radius fractures in daily life-A nonrandomized controlled trial in women

Kengo Usuki[1,2], Hiroaki Ueda[2], Toshiya Yamaguchi[3], Takako Suzuki[1], Toyohiro Hamaguchi[1]*

1 Department of Rehabilitation, Graduate School of Health Sciences, Saitama Prefectural University, Saitama, Japan, 2 Rehabilitation Center, Kitasato University Medical Center, Saitama, Japan, 3 Toda Chuo General Hospital, Honcho Toda, Saitama, Japan

* hamaguchi-toyohiro@spu.ac.jp

## Abstract

### Objective

Prolonged immobilization of joints after distal radius fracture (DRF) causes cerebral disuse-dependent plasticity (DDP) and deterioration of upper extremity function. Action observation therapy (AOT) can improve DDP.

### Trial design

This nonrandomized controlled trial (UMIN 000039973) tested the hypothesis that AOT improves hand-use difficulties during activities of daily living in patients with DRF.

### Method

Right-handed women with volar locking plate fixation for DRF were divided into AOT and Non-AOT groups for a 12-week intervention. The primary outcome was difficulty in using the fractured hand, assessed with the Japanese version of the Patient-related Wrist Evaluation (PRWE). The secondary outcomes were range of motion (ROM) of the injured side and gap between measured ROM and patient-estimated ROM. The survey was administered immediately post operation and at postoperative weeks 4, 8, and 12. The AOT group used a head-mounted display and three-dimensional video during ROM exercises. The Non-AOT group used active ROM exercises alone. A generalized linear model (GLM) was used to confirm interactions and main effects by group and time period, and multiple comparisons were performed.

### Results

Thirty-five patients were assigned to the AOT group (n = 18, median age, 74 years) or the Non-AOT group (n = 17, median age, 70 years). In the GLM, PRWE Total, PRWE Specific, and PRWE Usual scores revealed interactions between groups and periods. The post-hoc

**Data Availability Statement:** The data underlying the results presented in the study are available from URL https://osf.io/869kr/.

**Funding:** This work was supported by JSPS KAKENHI Grant Number 26750226 and 22K21243. The funders had no role in the study design, data collection and analysis, decision to publish, or preparation of the manuscript.

**Competing interests:** The authors have declared that no competing interests exist.

test revealed that the PRWE Specific scores ($z = 3.43$, $p = 0.02$) and PRWE Usual scores ($z = 7.53$, $p<0.01$) were significantly lower in the AOT group than in the Non-AOT group at 4 weeks postoperatively, whereas PRWE Total scores ($z = 3.29$, $p = 0.04$) were lower at 8 weeks postoperatively.

## Conclusions

These results suggested that AOT can improve hand-use difficulties in right-handed women after DRF surgery. AOT positively affects the motor imagery of patients with DRF and can reverse the patient's perceived difficulty in using the fractured hand during rehabilitation.

## Introduction

Distal radius fracture (DRF) is a common diagnosis, particularly prevalent in women [1, 2]. The DRF risk is 3.2 times higher in individuals who go outside at least once a day than those who do not [3–5]. This increased risk is often associated with activities related to household chores and occupation responsibilities. Rehabilitation following distal radius fracture surgery aims to restore hand usability early. However, it has been reported that there is no difference in the final outcome between rehabilitation and voluntary training [6, 7].

Range of motion (ROM) exercises are feasible in the early postoperative period for patients with volar locking plate fixation [8]. However, regaining the ability to use the fractured hand for activities of daily living (ADLs) takes time, typically around 9 weeks for bone union post-surgery [9, 10]. Although early loading after volar locking plate surgery is possible [11–13], patients commonly experience difficulties in hand use for ADLs at 8 weeks postoperatively, with a reduction in these difficulties by 50% at 12 weeks [14].

ROM limitation contributes to the challenges faced by patients with DRF in using their hands for ADLs (See S1 File) [15, 16]. The wrist and forearm ROMs used for ADLs have been reported as 38˚ for volar flexion, 40˚ for dorsiflexion, 13˚ for pronation, and 53˚ for supination [15]. Patients require 4 weeks postoperatively to recover to the level of ROM needed for ADLs. Additionally, dorsiflexion of the wrist joint at 4 and 8 weeks postoperatively has been identified as a contributing factor to perceived difficulty in hand use for ADLs [16].

Patients with DRF often underestimate their actual ROMs due to the immobilization of the joint [17], resulting in peripheral degeneration of sensory receptors [18, 19] and a decrease in superficial and positional sensation [20]. The immobilization causes the volume of the motor-sensory cortex of the brain and cerebral blood flow of the motor-sensory cortex to decrease [21, 22], eventually resulting in reduced motor sensation. Thus, even if the ROM improves to the angle required for performing ADLs, patients who underestimate their ROM may refrain from using their hands or subjectively experience difficulty in using them.

Decreased hand use causes a transient decrease in local brain activity related to hand movements, which, if persistent, alters the neural structure of the brain. This phenomenon is called disuse-dependent plasticity (DDP) [23]. If DDP occurs in patients with DRF and their ADLs are restricted by perceived difficulties in using their hands, then training to correct this phenomenon should be considered. Action observation therapy (AOT) may prevent DDP in patients. AOT takes advantage of the activation of the brain cortex when observing the actions of others [24]. Rocca et al. [25] conducted AOT of upper limb function in 42 healthy participants and reported that the AOT group, compared with the control group, had thicker gray matter of the cerebrum and improved upper limb function. AOT in patients with DRF may

prevent or reverse postoperative DDP and eliminate exacerbating factors of hand disuse. If the perceived difficulty is decreased, patients with DRF may be encouraged to use their hands for ADLs. No report exists regarding AOT for postoperative hand use in patients with DRF. This study tested the hypotheses that AOT in the early postoperative period of DRF can (1) prevent the underestimation of ROM, (2) attenuate hand-use difficulty in ADL, and (3) improve joint ROM early in patients with DRF (S2 File).

## Methods

### Study design

This study was a nonrandomized controlled trial conducted between 2013.10.1 and 2021.6.1. The data collection was concluded because the planned number of participants was reached.

### Ethical considerations

Written informed consent was obtained from all patients. The study was registered as a clinical trial (UMIN 000039973) and approved by the Ethics Committees of Saitama Prefectural University (Saitama, Japan; approval no. 25512, 27017) and Kitasato University Medical Center (Saitama, Japan; approval no. 27–57).

### Participants

The eligibility criteria for this study were women with DRF who underwent volar locking plate fixation during the study period. The exclusion criteria were as follows: (1) patients who did not wish to participate, (2) patients under 18 years of age, (3) patients with substantial soft tissue injuries in addition to fractures, (4) patients with fractures due to tumors, (5) patients with rheumatoid disorders, (6) patients with neurological diseases, and (7) patients with cognitive dysfunction. The data collection was conducted at the Kitasato University Medical Center during the rehabilitation.

### Experimental procedures

Study participants were assigned to one of two groups: the group that received AOT and the group that received the usual rehabilitation without AOT (Non-AOT). The allocation was not randomized: 10 patients were assigned to the Non-AOT group from September 2013 to August 2014 and to the AOT group from January 2016 to October 2018 to adjust the experimental equipment (i.e., Ghost), depending on the time of the year. The patients were thereafter assigned to the two groups alternately in the order of their prescriptions until the sample size was met in March 2021.

### Evaluation method

In this study, the following 11 items were investigated: (1) age; (2) dominant hand (based on the Edinburgh Handedness Inventory); (3) time from injury to start of rehabilitation (days); (4) fracture severity, based on the Arbeitsgemeinschaft für Osteosynthesefragen (AO) classification; (5) body mass index; (6) injured side; (7) healthy side active ROM; (8) affected side active ROM; (9) patient's ROM estimate for the affected side in relation to the maximum wrist ROM of the healthy side (%); (10) patient's estimated active ROM (˚), and (11) difficulty in using the hand for ADLs, based on the PRWE. The main outcome was the PRWE and the secondary were actual and estimated ROM. The survey was conducted at the time of initial rehabilitation, 4 weeks postoperatively, 8 weeks postoperatively, and 12 weeks postoperatively. The reason for this survey timing is that patients with DRF have temporary bone formation at 3–4

weeks postoperatively, whereas the fracture site stabilizes at 6–8 weeks postoperatively, and the orthopedic surgeon gives permission for patients to perform ADLs without restrictions. In addition, rehabilitation is completed at 12 weeks [26].

S3 File shows the method of measuring the gap between the patient's estimated ROM (˚) and the measured ROM (˚) of the affected side, based on the maximum active ROM (100%) on the healthy side, as described previously [17]. The patient was instructed to (1) "rest the hand on the table and close your eyes" and (2) "estimate what percentage your hand moves relative to your healthy hand, assuming the healthy hand is 100%." The reason patients were asked to estimate the angle percentage in comparison to the healthy side was that they had difficulty determining absolute angles and to prevent them from visually confirming their answers during the estimation. Test runs were not performed because underestimation may be corrected in multiple tests.

Afterward, the patient's estimated ROM of the affected side (%) was converted into the estimated ROM (˚) of the affected side using Eq (1):

$$Estimated\ ROM\ (^{\circ}) = \frac{x}{100} \times the\ measured\ ROM\ on\ the\ healthy\ side \qquad \text{Eq (1)}$$

in which $x$ represents the patient's estimated ROM (%). The gap between the patient's estimated ROM (˚) and the measured ROM (˚) of the affected side was then calculated, using Eq (2):

$$\begin{aligned} Perceived\ &ROM\ gap\ (^{\circ}) \\ &= estimated\ ROM\ of\ the\ affected\ side\ (^{\circ}) - measured\ ROM\ of\ the\ affected\ side\ (^{\circ}) \quad \text{Eq (2)} \end{aligned}$$

In this study, the investigators were not blinded to the treatment methods.

## Intervention method

The intervention was conducted for 40 minutes per session, at least once a week, for 12 weeks after surgery. The intervention consisted of 5 minutes of evaluation, 10 minutes of passive ROM practice, another 10 minutes of active ROM practice, 10 minutes of upper limb function (i.e., ADL) practice, and 5 minutes of icing. Participants in the AOT group performed active ROM practice with the addition of AOT by using a head-mounted display (HMD) and first-person three-dimensional (3D) video. AOT can consist of first-person and third-person action observations, but the first-person motion illusion is easier to produce than is the third-person motion illusion, and the sense of bodily ownership of the hand on the screen is higher, as evidenced by the activity of the cerebral motor-related regions in near-infrared spectroscopy [27]. Thus, we employed an HMD-based first-person action observation system [28]. Participants in the Non-AOT group performed active ROM exercises without AOT. Interventions were conducted by physical or occupational therapists with at least 3 years of experience. S4 File shows the posture and the equipment used during the action observation using 3D moving images. We chose 3D moving images because brain activity in response to 3D moving images is higher than that of 2D moving images when the participant visualizes the movement after action observation [29].

AOT utilizes an upper limb motor function learning device (Code name: Ghost; patent number 6425335; Saitama Prefectural University, Saitama, Japan) [28]. Ghost comprises an HMD with personal computer software. A 3D image of another person's hand, which had been set in advance, was presented to the patient by the HMD. The presented videos were finger flexion and extension (20 seconds), volar flexion and dorsiflexion (5 minutes), and pronation and supination (5 minutes). Patients were not instructed to perform active motor imagery during AOT.

## Sample size calculation

The sample size of this study was calculated by repeated measures analysis of variance (i.e., Cohen effect size) for group and time period, following a previous study [30], with the difference in PRWE scores as the primary outcome, after modeling the interaction between two PRWE groups and four time periods as the pretest using a generalized linear model (GLM; $f = 0.25$, error $\alpha = 0.05$, $\beta = 0.80$, correlation among repeated measures = 0.5, non-sphericity correction = 1). The total number of cases was calculated as 24, with 12 in each group. The number of participants needed to prematurely terminate the study was estimated as 20%, and the final target sample size was set as 34 patients in total, with 17 cases in each group. G*Power software (v 3.1.9.6; Universität Kiel, Kiel, Germany) was used for these calculations [31].

## Statistical analysis

The primary outcome of this study was the PRWE score as a measure of the difficulty in using the hands for ADLs [32]. The secondary outcome was the gap between the estimated and measured ROM of the affected side.

The PRWE is a patient-oriented assessment of pain, difficulty in using a hand, and difficulty in using it in ADLs specific to disorders of the wrist joint. It consists of 15 questions, and the total score ranges from 0 to 100. A higher score indicates a higher degree of disability, and a score of 0 indicates no disability at all [33]. The reliability and validity of the Japanese version have been proven [34].

The Levene's test was used for the test of equal variances for all datasets. The Shapiro–Wilk test was used to test for normality. Descriptive statistics of the characteristics of the AOT and Non-AOT groups and the primary and secondary outcomes at baseline were compared by using the Mann–Whitney $U$ test. For the primary and secondary outcomes, we used the GLM to obtain the goodness of fit of the estimated models by group and time period. When the outcome values fitted the model, multiple comparisons were tested using the Bonferroni test, after checking the interaction between group and time. Age [35], body mass index [36], side of injury [37], fracture severity [38], and number of days between injury and start of rehabilitation [39] have been associated with functional outcomes, and these were used as the covariates. Outliers, defined as values exceeding the first quartile–quartile range × 1.5 or as the third quartile + quartile range × 1.5, were excluded from statistical analyses [40]. Missing values of measurements were assigned the median value of the group and time period [41]. Jamovi (version 1.6.16.0) was used for data analysis [42]. The significance level was set at 0.05.

## Results

### Patient registration and analysis process

Fig 1 shows the flowchart of the participant enrollment. A total of 89 patients with DRF presented during the study period, and 46 patients met the study selection criteria. A total of 11 patients were excluded from the statistical analysis because they discontinued their participation and were unable to attend the outpatient clinic. Therefore, data from 35 patients (18 in the AOT group and 17 in the Non-AOT group) were statistically analyzed. In all patients, a fall had caused the injury. An outlier test was performed before further statistical analyses, and the excluded data were 14 in the AOT group and 17 in the Non-AOT group for the measured ROM of the injured side (4 types, 4 time periods), 9 in the AOT group and 13 in the Non-AOT group for the gap between estimated and measured ROMs (4 types, 4 time periods), and 9 in the Non-AOT group for the PRWE score (4 types, 4 time periods). Table 1 shows the characteristics of the AOT and Non-AOT groups. The characteristics did not differ between the

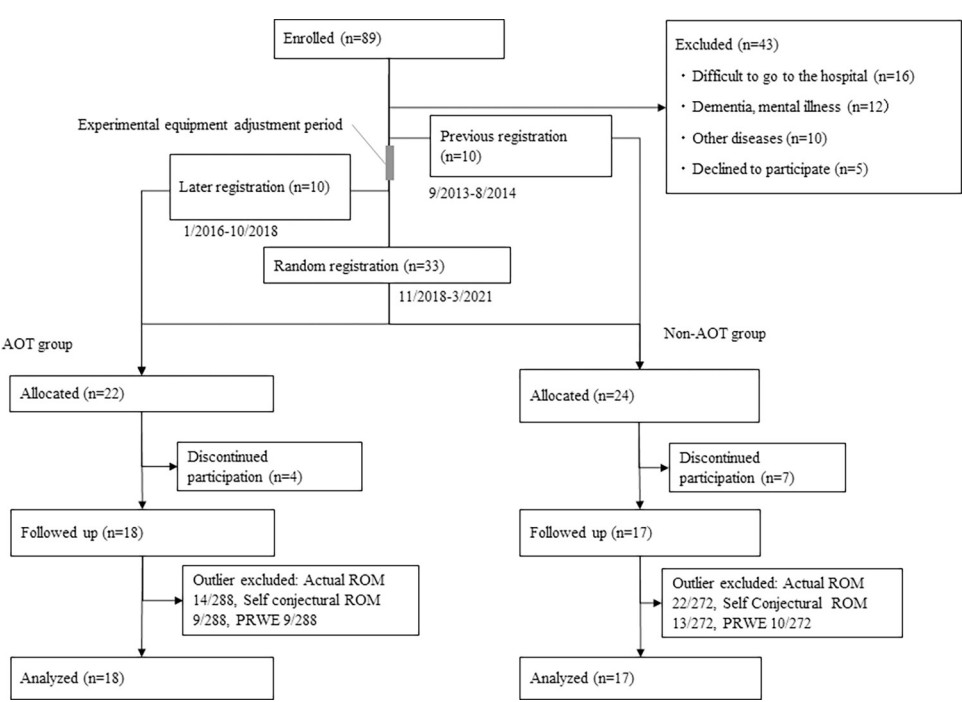

**Fig 1. Flowchart of the participants' enrollment.** During the study period, 89 postoperative DRF patients were admitted to the hospital, and 46 of them met the eligibility criteria. Of these, 11 patients were excluded because they were no longer able to attend the outpatient clinic. Finally, 35 patients were followed up, and their data were statistically analyzed. AOT, action observation therapy; DRF, distal radius fracture; HMD, head-mounted display; PRWE, Patient-related Wrist Evaluation; ROM, range of motion.

AOT and Non-AOT groups. Two patients in the AOT group and seven patients in the Non-AOT group underwent surgery using fixation materials such as a long plate, percutaneous pinning, and an ulnar plate other than the usual volar locking plate. At the time of evaluation, one patient in the AOT group and two patients in the Non-AOT group were diagnosed as having incomplete bone union.

**Table 1. Characteristics of the study participants.**

| Characteristic | Group | | Statistics | |
|---|---|---|---|---|
| | AOT group (n = 18) | Non-AOT group(n = 17) | U | p value |
| Age, y | 74 (66, 76) | 70 (60, 78) | 142 | 0.72 |
| Body mass index | 22 (19, 25) | 23 (21, 28) | 115 | 0.22 |
| Handedness, right | 100 (100, 100) | 100 (100, 100) | - | - |
| Injured hand, right | 9 | 5 | 113 | 0.13 |
| Start of rehabilitation after injury (day) | 11 (9, 17) | 10 (9, 15) | 139 | 0.64 |
| Fracture severity | 3 (2, 3) | 3 (2, 3) | 143 | 0.71 |
| Extra-articular (A) | 0 | 1 | - | - |
| Partially articular (B) | 7 | 5 | - | - |
| Complete articular (C) | 11 | 11 | - | - |
| Number of rehabilitation sessions | 14 (10, 19) | 17 (15, 24) | 105 | 0.11 |

Data are presented as the median and quartile range (25%, 75%). Fracture severity is based on the Arbeitsgemeinschaft für Osteosynthesefragen (AO) classification.

Handedness is based on the Edinburgh Handedness Inventory. All participants were right-handed.

AOT, action observation therapy

**Table 2. Comparisons of the gap between the conjectural and actual range of motion among the groups at baseline.**

| | Group | | Statistics | |
|---|---|---|---|---|
| | AOT group (n = 18) | Non-AOT group (n = 17) | U | p value |
| VF_Actual_Healthy-side | 73 (70, 80) | 65 (60, 80) | 104 | 0.10 |
| DF_Actual_Healthy-side | 75 (70, 80) | 70 (70, 77) | 129 | 0.60 |
| Pronation_Actual_Healthy_side | 90 (90, 90) | 90 (90, 90) | - | - |
| Supination_Actual_Healty_side | 90 (90, 90) | 90 (90, 90) | - | - |
| VF_Actual_Affected_side | 40 (30, 43) | 35 (25, 40) | 97 | 0.24 |
| DF_Actual_Affected_side | 38 (20, 40) | 30 (25, 40) | 130 | 0.44 |
| Pronation_Actual_Affected_side | 60 (46, 75) | 50 (38, 55) | 84 | 0.04 |
| Supination_Actual_Affected_side | 55 (40, 78) | 50 (40, 70) | 150 | 0.92 |
| VF_Conjectual_Affected_side | 24 (8, 30) | 16 (12, 24) | 138 | 0.10 |
| DF_Conjectual_Affected_side | 8 (4, 22) | 14 (8, 35) | 122 | 0.20 |
| Pronation_Conjectual_Affected_side | 27 (6, 52) | 27 (9, 45) | 145 | 0.05 |
| Supination_Conjectual_Affected_side | 27 (8, 61) | 9 (9, 45) | 123 | 0.20 |
| VF_Gap | -13 (-28, -2) | -20 (-23, -10) | 135 | 0.77 |
| DF_Gap | -20 (-29, -11) | -13 (-22, 0) | 113 | 0.19 |
| Pronation_Gap | -15 (-33 to -8) | -23 (-38, 5) | 145 | 1.00 |
| Supination_Gap | -16 (-30, -9) | -28 (-50, -12) | 99 | 0.18 |
| PRWE_Total | 73 (56, 79) | 70 (61, 78) | 124 | 0.89 |
| PRWE_Pain | 29 (19, 33) | 27 (26, 30) | 63.5 | 0.74 |
| PRWE_Specific | 60 (57, 60) | 57 (49, 60) | 142 | 0.06 |
| PRWE_Usual | 31 (20, 38) | 32 (23, 35) | 125 | 0.92 |

Data are presented as the median and quartile range (25%, 75%). Pronation_Actual_Healthy_side scores and Supination_Actual_Heathy_side scores are all 90, without variation.

AOT, action observation therapy; DF, dorsal flexion; VF, volar flexion

## PRWE and ROM baseline data

Table 2 shows the baseline values of the measured ROM on the healthy side, measured ROM of the affected side, estimated ROM of the affected side, gap between the estimated and measured ROM of the affected side, and PRWE score in the AOT and Non-AOT groups for the purpose of showing that no change existed between the AOT group and the Non-AOT group at baseline. The values for the measured ROM on the healthy side, measured ROM of the affected side, estimated ROM of the affected side, gap between estimated ROM and measured ROM of the affected side, and PRWE scores were not significantly different between the AOT and Non-AOT groups at baseline. The normality of the baseline data was evaluated using the Shapiro–Wilk test. The PRWE Pain score (W = 0.97, $p$ = 0.41) and PRWE Total score (W = 0.96, $p$ = 0.19) had a normal distribution, whereas the PRWE Specific score (W = 0.75, $p < 0.01$) and PRWE Usual score (W = 0.90, $p < 0.01$) did not have a normal distribution. Therefore, the tests for PRWE, ROM, and gap between estimated and measured ROMs were conducted after the GLM fit was confirmed.

## Variation of PRWE and ROM by group and time period: GLM fit

Table 3 shows the changes in the measured ROM, estimated ROM, gap between the estimated and measured ROMs, and PRWE scores of the AOT and Non-AOT groups over time for the purpose of showing changes in each inspection item over time. Disability decreased in all test items from baseline to 3 months after surgery. Table 4 shows the comparative results of the

**Table 3. Group comparisons for actual ROM, conjectural ROM, gap between the actual and conjectural ROMs, and PRWE score.**

| | Baseline | | Postop. 4W | | Postop. 8W | | Postop. 12W | |
|---|---|---|---|---|---|---|---|---|
| | AOT (n = 18) | Non-AOT (n = 17) | AOT (n = 18) | Non-AOT (n = 17) | AOT (n = 18) | Non-AOT (n = 17) | AOT (n = 18) | Non-AOT (n = 17) |
| VF_Actual_Affected_side | 40 (30, 43) | 35 (25, 40) | 55 (50, 60) | 48 (40, 55) | 60 (56, 65) | 50 (45, 60) | 63 (56, 70) | 60 (50, 60) |
| DF_Actual_Affected_side | 38 (20, 40) | 30 (25, 40) | 58 (54, 60) | 55 (48, 60) | 65 (60, 70) | 60 (50, 60) | 65 (60, 70) | 60 (60, 60) |
| Pronation_Actual_Affected_side | 60 (46, 75) | 50 (38, 55) | 90 (73, 90) | 80 (75, 90) | 90 (90, 90) | 80 (70, 90) | 90 (90, 90) | 90 (70, 90) |
| Supination_Actual_Affected_side | 55 (40, 78) | 50 (40, 70) | 90 (85, 90) | 80 (60, 90) | 90 (90, 90) | 90 (86, 90) | 90 (90, 90) | 90 (90, 90) |
| VF_Conjectual_Affected_side | 24 (8, 30) | 16 (12, 24) | 48 (35, 59) | 48 (36, 52) | 54 (48, 56) | 49 (42, 56) | 56 (48, 67) | 56 (48, 65) |
| DF_Conjectual_Affected_side | 8 (4, 22) | 14 (8, 35) | 43 (29, 56) | 38 (24, 49) | 51 (41, 72) | 48 (42, 56) | 56 (44, 63) | 56 (49, 63) |
| Pronation_Conjectual_Affected_side | 27 (6, 52) | 27 (9, 45) | 77 (45, 89) | 63 (45, 72) | 81 (72, 90) | 72 (45, 81) | 86 (81, 90) | 86 (72, 90) |
| Supination_Conjectual_Affected_side | 27 (8, 61) | 9 (9, 45) | 63 (48, 84) | 68 (54, 72) | 81 (72, 90) | 81 (72, 90) | 81 (72, 90) | 81 (63, 90) |
| VF_Gap | -13 (-28, -2) | -20 (-23, -10) | -8 (-13, -2) | -4 (-6, 2) | -10 (-17, -2) | 0 (-8, 5) | -2 (-14, 5) | 1 (-6, 10) |
| DF_Gap | -20 (-29, -11) | -13 (-22, 0) | -16 (-24, -5) | -11 (-21, -9) | -10 (-24, 0) | -11 (-17, 0) | -13 (-19, 2) | -2 (-4, 0) |
| Pronation_Gap | -15 (-33, -8) | -23 (-38, 5) | -9 (-27, 0) | -16 (-18, -9) | 0 (-18, 0) | -17 (-26, -9) | 0 (-9, 0) | 0 (-9, 0) |
| Supination_Gap | -16 (-30, -9) | -28 (-50, -12) | -18 (-31, 0) | -8 (-20, 0) | -9 (-17, 0) | -9 (-18, 0) | -9 (-17, 0) | 0 (-12, 0) |
| PRWE_Total | 73 (56, 79) | 70 (61, 78) | 38 (27, 45) | 50 (37, 68) | 20 (11, 28) | 37 (28, 54) | 14 (6, 23) | 18 (8, 37) |
| PRWE_Pain | 29 (19, 33) | 27 (26, 30) | 17 (11, 21) | 24 (13, 28) | 9 (6, 15) | 21 (9, 26) | 7 (3, 11) | 7 (4, 18) |
| PRWE_Specific | 60 (57, 60) | 57 (49, 60) | 28 (21, 32) | 40 (32, 52) | 14 (7, 20) | 26 (22, 32) | 9 (1, 16) | 11 (4, 27) |
| PRWE_Usual | 31 (20, 38) | 32 (23, 35) | 10 (7, 14) | 24 (23, 26) | 4 (3, 8) | 10 (6, 23) | 4 (1, 6) | 1 (0, 10) |

Data are presented as the median and quartile range (25%, 75%).

AOT, action observation therapy; DF, dorsal flexion; postop., postoperative; PRWE, Patient-related Wrist Evaluation; ROM, range of motion; VF, volar flexion; W, week

ROM of the affected side, gap between estimated and measured ROMs, and PRWE scores of the AOT and Non-AOT groups by GLM for the purpose of clarifying the effects of AOT. Fig 2 shows the comparison of the temporal PRWE changes between the AOT and Non-AOT groups.

**Table 4. Comparisons of the gap between the conjectural and actual range of motion between the study groups.**

| | Model Information | | Model result | | Effect size | 95% CI | Statistics | |
|---|---|---|---|---|---|---|---|---|
| | $R^2$ | AIC | $\chi^2$ | p | Exp (B) | Lower to Upper | z | p |
| PRWE_Total | 0.66 | 1164 | 8.10 | 0.04 | <0.01 | -31.54 to -2.76 | -2.34 | 0.02 |
| PRWE_Pain | 0.45 | 979 | 2.38 | 0.50 | 0.08 | -10.72 to 5.55 | -6.2 | 0.54 |
| PRWE_Specific | 0.73 | 1038 | 16.79 | <0.01 | <0.01 | -31.31 to -10.19 | -3.68 | <0.01 |
| PRWE_Usual | 0.68 | 931 | 14.63 | <0.01 | <0.01 | -21.14 to -8.52 | -3.61 | <0.01 |
| VF_Actual_Affected_side | 0.58 | 998 | 2.35 | 0.49 | 293.19 | -3.49 to 15.08 | 1.20 | 0.23 |
| DF_Actual_Affected_side | 0.71 | 975 | 2.12 | 0.55 | 3.05 | -7.64 to 9.53 | 0.26 | 0.8 |
| Pronation_Actual_Affected_side | 0.53 | 1110 | 2.35 | 0.50 | <0.01 | -21.13 to 4.96 | -1.21 | 0.23 |
| Supination_Actual_Affected_side | 0.54 | 1082 | 4.02 | 0.26 | 64193.96 | -2.7 to 25.10 | 1.61 | 0.11 |
| VF_Gap | 0.27 | 1058 | 6.82 | 0.08 | <0.01 | -23.45 to 0.30 | -1.95 | 0.05 |
| DF_Gap | 0.13 | 1100 | 1.58 | 0.67 | 429.34 | -6.24 to 18.56 | 0.97 | 0.34 |
| Pronation_Gap | 0.17 | 1153 | 1.85 | 0.6 | 4.833 | -16.04 to 18.95 | 1.25 | 0.21 |
| Supination_Gap | 0.23 | 1183 | 7.76 | 0.04 | <0.01 | -38.22 to -4.04 | -2.49 | 0.02 |

Data are presented as the average value and quartile range (25%, 75%). The distribution is Gaussian.

AIC, Akaike Information Criterion; CI, confidence interval; DF, dorsal flexion; VF, volar flexion

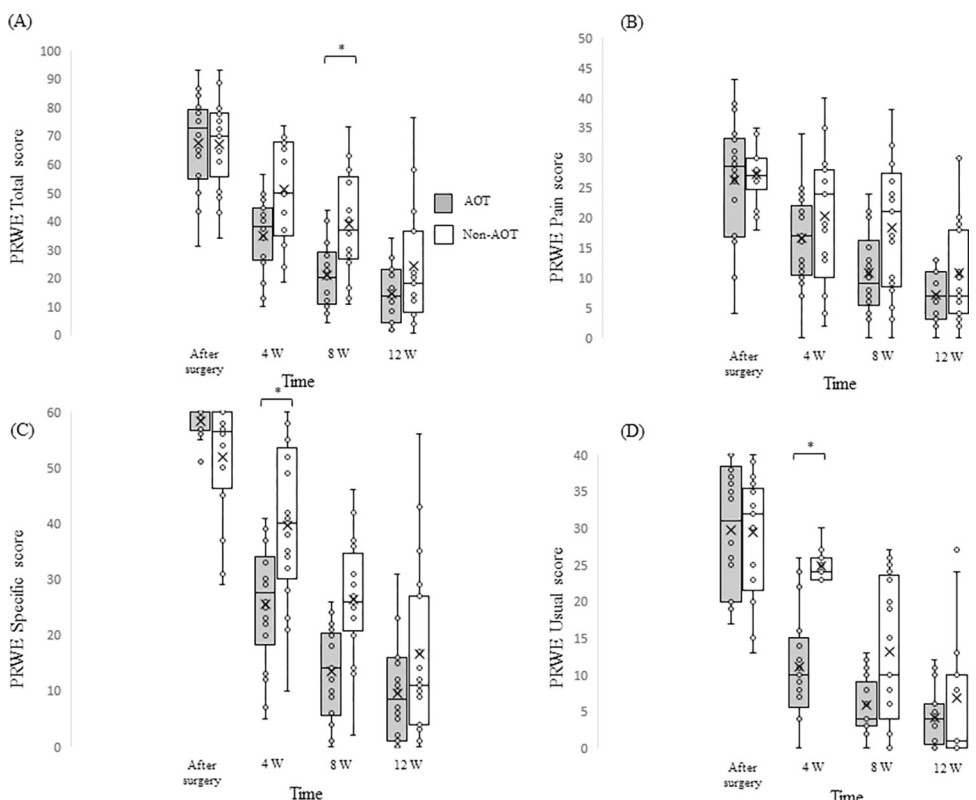

**Fig 2. Comparison of PRWE scores between the AOT and Non-AOT groups.** (A) PRWE Total score was significantly lower in the AOT group than in the Non-AOT group at 8weeks postoperatively. (B) PRWE Pain scores did not differ between the AOT and Non-AOT groups. (C, D) PRWE Specific and Usual score was significantly lower in the AOT group than in the Non-AOT group at 4weeks postoperatively *Bonferroni, post-hoc test after GLM fitting, $p<0.05$. AOT, action observation therapy; GLM, generalized linear model; PRWE, Patient-related Wrist Evaluation; W, week.

### Differences in PRWE scores between the AOT and Non-AOT groups

PRWE Total ($R^2$ = 0.66, Akaike Information Criterion [AIC] = 1164, $\chi^2$ = 8.10, $p$ = 0.04), PRWE Specific ($R^2$ = 0.73, AIC = 1038, $\chi^2$ = 16.79, $p<0.01$), and PRWE Usual ($R^2$ = 0.68, AIC = 931, $\chi^2$ = 14.63, $p<0.01$) showed a significant interaction between group and time. No interaction existed between group and time for PRWE Pain ($R^2$ = 0.45, AIC = 979, $\chi^2$ = 2.38, $p$ = 0.50).

Bonferroni post-hoc test results (presented as the median [25%, 75% quartile]) showed that the PRWE Total score was 20 [11, 28] in the AOT group and 37 [28, 54] in the Non-AOT group at 8 weeks (Z = 3.29, $p$ = 0.04); the PRWE Specific score was 28 [21, 32] in the AOT group and 40 [32, 52] in the Non-AOT group at 4 weeks (Z = 3.42, $p$ = 0.02); and the PRWE Usual score was 20 [7, 14] in the AOT group and 24 [23, 26] in the Non-AOT group at 4 weeks (Z = 3.42, $p$ = 0.02).

### Differences in the measured ROM of the affected side between the AOT and Non-AOT groups

The active ROMs of the affected side differed for volar flexion ($R^2$ = 0.58, AIC = 998, $\chi^2$ = 2.35, $p$ = 0.49), dorsiflexion ($R^2$ = 0.71, AIC = 975, $\chi^2$ = 2.12, $p$ = 0.49), pronation ($R^2$ = 0.53, AIC = 1110, $\chi^2$ = 2.35, $p$ = 0.50), and supination ($R^2$ = 0.54, AIC = 1182, $\chi^2$ = 4.02, $p$ = 0.26), but without a significant difference, between the AOT and Non-AOT groups.

### The gap between the estimated ROM and measured ROM in the AOT and Non-AOT groups

With regard to the gap between the estimated and measured ROM of the affected side, supination ($R^2$ = 0.23, AIC = 1183, $\chi^2$ = 7.76, $p$ = 0.04) showed a significant interaction between group and time. For volar flexion ($R^2$ = 0.27, AIC = 1058, $\chi^2$ = 6.82, $p$ = 0.08), dorsal flexion ($R^2$ = 0.13, AIC = 1100, $\chi^2$ = 1.58, $p$ = 0.67), and pronation ($R^2$ = 0.17, AIC = 1153, $\chi^2$ = 1.85, $p$ = 0.60), no difference existed between the AOT and Non-AOT groups. The Bonferroni post-hoc test showed that supination was not significantly different at any group and time.

### Sensitivity analysis to outlier

PRWE Specific ($R^2$ = 0.67, AIC = 1102, $\chi^2$ = 14.28, $p<0.01$) showed a significant interaction between group and time. No interaction existed between group and time for PRWE Pain ($R^2$ = 0.43, AIC = 1025, $\chi^2$ = 2.25, $p$ = 0.52), PRWE Usual ($R^2$ = 0.57, AIC = 1011, $\chi^2$ = 6.77, $p$ = 0.08), and PRWE Total ($R^2$ = 0.65, AIC = 1179, $\chi^2$ = 6.95, $p$ = 0.07).

## Discussion

The results of this study suggest that, when right-handed women with DRF receive AOT using first-person 3D video immediately after surgery, their perceived difficulty in using the affected hand in daily life and their sense of difficulty in daily life are significantly improved at 4 weeks postoperatively, compared with that of patients without AOT. Since AOT can maintain the sense of use of the hand by affecting the motor imagery of patients with DRF, it may promote the use of this hand when performing ADLs during postoperative rehabilitation after DRF surgery.

The results of neurological studies in which AOT improved the perceived difficulty of using the injured hand in ADLs suggest that AOT increases neural excitation from the primary motor cortex to muscles via corticospinal tracts [43] and prevents cortical DDP [25], even without actual movement. Rocca et al. [25] administered AOT for upper limb function in 42 healthy participants and reported that, compared with the control group, the AOT group had improved thickened gray matter of the cerebrum and upper limb function. Fadiga et al. [43] compared the motor-evoked potentials of the extensor digitorum communis, flexor digitorum superficialis, first dorsal interosseus, and opponens pollicis in 12 healthy individuals when they observed someone holding and reaching for an object, as well as when they observed a stationary object. Motor-evoked potentials did not increase during the observation of a stationary object but during the observation of a hand releasing or reaching for an object [43]. These studies support the neurophysiological point of view that AOT is effective in preventing DDP, even in the early postoperative period when pain is strong and the patient is unable to use the affected hand because bone fragility limits its use in ADLs.

Opie et al. [44] restrained the dominant index finger of healthy participants for 8 hours and found a decrease in cortical excitability and a decrease in fine hand movements, whereas motor skill learning was similar to that of participants without 8 hours of restraint. The improvement in cortical excitability and fine hand movements was greater after immobilization [44]. van de Ruit and Grey [45] also investigated cortical excitability after visual motor learning and reported that the area of transcranial magnetic stimulation-induced upper limb movements increased by 18%–36% after motor learning, which indicated that brain plasticity had occurred [45]. DDP is a phenomenon in which cortical excitability is reduced by fixation or disuse [21, 22], whereas use-dependent plasticity is a phenomenon in which cortical excitability is increased by upper limb function practice and motor learning using vision, thereby

resulting in plasticity. We assumed that the Ghost-based action observation and active ROM exercises of the hand joints were effective through use-dependent plasticity, thereby preventing DDP.

In the present study, the patients' difficulties in using the affected hand for ADLs had improved at 4 weeks postoperatively, which suggested that AOT intervention effectively lowered the perceived difficulty in using the hand for ADLs and social activities from this time-point onward. Dilek et al. [46] reported improvement in pain, ROM, and ADL (assessed with the Michigan Hand Questionnaire and the Disability Arm Shoulder and Hand) at 8 weeks after DRF surgery, after patients underwent motor imagery tasks and the usual rehabilitation program. Thus, AOT is effective as an early postoperative intervention without exercise in the early postoperative period after DRF surgery.

MacDermid et al. [14] reported that the median PRWE Total scores after DRF surgery changed over time, with a score of 43.5 at 8 weeks postoperatively. The AOT group in the present study had a median PRWE Total score of 21 at 8 weeks, which suggested an improvement, compared with the results of the aforementioned study. However, the ROM underestimation was not reduced by AOT, indicating that AOT affected the perception of hand use but does not affect the actual ROM and its estimated value.

One reason AOT did not significantly improve ROM underestimation in patients with DRF is the influence of surgical intervention. Research indicates that the flexor pollicis longus tendon, which is elongated during dorsiflexion, is impaired after volar locking plate surgery [47]. Regarding forearm rotation, the pronator quadratus is located at the site of the surgical wound, and the ROM may have been influenced by the surgical manipulation.

DDP in patients with DRF may also be influenced by changes in peripheral sensation after a fracture. Karagiannopoulos et al. [20] reported that the hands of patients with DRF had decreased superficial sensation and proprioception, compared with healthy hands. Michinaka et al. [19] reported a decreased number of sensory receptors with typical morphology and an increased number of sensory receptors with atypical morphology in the anterior cruciate ligament after 6 weeks of knee joint immobilization in rabbits. After 24 weeks of subsequent mobilization, these numbers were not significantly different from their baseline values, although the number of atypical nerve endings remained high. Nencini and Ivanusic [48] found that the periosteum and bone marrow contain many receptors for mechanical stimuli, receptors for chemical stimuli, and receptors for pain and heat, which are involved in somatosensory perception. If types and quantities of sensory receptors are similarly altered in patients with DRF, an implication is that sensory information from the periphery is mistakenly processed in the brain, possibly causing DDP.

Recovery after DRF is divided into the inflammatory phase, the bone fusion repair phase, and the remodeling phase. Marsell and Einhorn [49] state that the remodeling phase begins at 3 to 4 weeks in humans and in animal models and may take up to several years. During the 12-week period investigated in the present study, the peripheral nerve endings and the periosteum of tissues related to somatosensory perception were not completely repaired and remained impaired, which suggests that AOT had little effect on the underestimation. AOT did not influence the measured ROM of the affected side. Brehmer and Husband [50] compared 81 postoperative patients with volar locking plate fixation. They were divided into two groups: one group started passive ROM and muscle strengthening exercises at 2 weeks postoperatively and the other group, at 6 weeks postoperatively. Compared with the group that started at 6 weeks postoperatively, the group that started at 2 weeks had improved ROM, muscle strength, and ADLs (based on the Disability Arm Shoulder and Hand score) at 8 weeks [50]. In the present study, no difference existed between the AOT and Non-AOT groups with

regard to ROM and muscle strength with AOT alone, which suggested lack of an intervention effect on the actual measured ROM in the affected hand.

Previous research has suggested that a biosocial psychological model is involved in recovery after distal radius fracture surgery [51, 52]. Steven et al. state that high or increasing levels of catastrophizing had an increased risk for a less than full recovery of strength by almost six-fold [51]. It is unclear what characteristics the participants had in terms of biosocial psychology, and what kind of influence AOT had on them.

In the sensitivity analyses that included data outliers, we found that the significant differences in PRWE Total and Usual scores previously observed were no longer present. This suggests that our earlier results might be sensitive to extreme values. However, the significant difference in PRWE Specific scores remained, indicating a persistent effect of the AOT intervention on specific functional improvements. These findings have been discussed in the context of their implications for clinical practice, particularly in settings where outlier data cannot be readily excluded [53, 54]. This study showed that AOT has the potential to improve the feeling of using the hand in daily life, but further research with a larger number of participants is needed to prove its effectiveness.

## Study limitations

This study was a nonrandomized controlled trial. Therefore, information bias cannot be completely eliminated. The intervention was successful in older, right-handed women; whether it is equally effective in younger women or men or in left-handed patients needs to be examined separately.

Compared with the AOT group, more patients in the Non-AOT group received fixation materials other than volar locking plate fixation. Additionally, there was one non-AOT patient who was considered to have incomplete bone union at the 3-month evaluation, which may have affected the results. The surgical procedure and the delay in bone union are likely to have influenced the results.

Pain and grip strength are factors affecting ADLs after DRF, based on the results of a study in patients immobilized for 6 weeks with K-wire from weeks 6 to 24 [55]. Previous studies report differences in sensation and grip strength; however, the present study did not take patient-specific weight load into consideration when collecting data. Based on the guidelines, maximal grip strength is not recommended until 4 weeks. In the pre-experiment, sensation was not different between fractured and nonfractured hands, indicating that sensation was not a problem. Motor imagery abilities are known to decrease in older individuals [56, 57].

The elderly may not be able to adapt to the changes in motor function caused by a DRF because of their reduced motor imagery ability. The approach in this study may have facilitated improvement. Moreover, whether AOT induced an improvement in ADLs with the affected hand was because of motor imagery effects or because of actual movements remains unclear. The difference between the effects of 2D and 3D images was also not clarified in this study. Furthermore, differences in AOT effects in relation to cognitive function, mental function, fracture severity, and the presence or absence of periosteal damage were not clarified and should be examined in future studies.

## Conclusions

In addition to the usual rehabilitation, utilizing AOT with first-person 3D moving images in postoperative women with DRF can improve the perceived difficulty in using hands for ADL and daily life at 4 weeks and 8 weeks postoperatively. This study was not a randomized

controlled trial; therefore, the efficacy of this intervention must be verified in follow-up studies, including whether it is effective, regardless of age and sex.

## Supporting information

**S1 File. Range of motion of the wrist joint required for activities of daily living.**
(PDF)

**S2 File. Research hypotheses.**
(PDF)

**S3 File. Gap between the estimated and measured ROM of the affected side in wrist flexion.**
(PDF)

**S4 File. Patient posture during the Ghost intervention.**
(PDF)

**S5 File. Study protocol.**
(PDF)

**S6 File. Raw experimental data.**
(CSV)

**S1 Video. Experimental process video.**
(MOV)

## Acknowledgments

The authors would like to thank Associate Professor Toshiyuki Ishioka and Professor Kenichi Tanaka of the Graduate School of Saitama Prefectural University (Saitama, Japan) for their guidance in the preparation of this paper. We thank the staff of the Department of Orthopedics, Kitasato University Medical Center, the Department of Physical Therapy and Occupational Therapy, Rehabilitation Center, and Toda Central Rehabilitation Clinic for their cooperation in this study. This work was supported by a Japan Society for the Promotion of Science (JSPS) KAKENHI Grant (grant numbers 26750226 and 22K21243).

## Author Contributions

**Conceptualization:** Takako Suzuki, Toyohiro Hamaguchi.

**Data curation:** Kengo Usuki, Hiroaki Ueda, Toshiya Yamaguchi.

**Formal analysis:** Kengo Usuki.

**Funding acquisition:** Kengo Usuki, Takako Suzuki.

**Investigation:** Kengo Usuki, Hiroaki Ueda, Toshiya Yamaguchi.

**Methodology:** Kengo Usuki, Toyohiro Hamaguchi.

**Project administration:** Kengo Usuki, Toyohiro Hamaguchi.

**Supervision:** Takako Suzuki, Toyohiro Hamaguchi.

**Validation:** Hiroaki Ueda, Toshiya Yamaguchi, Takako Suzuki.

**Visualization:** Kengo Usuki, Hiroaki Ueda, Toshiya Yamaguchi.

**Writing – original draft:** Kengo Usuki, Toyohiro Hamaguchi.

**Writing – review & editing:** Toyohiro Hamaguchi.

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
