## [Decision Letter · Decision Letter 0]

29 Feb 2024

PONE-D-23-33804Action observation intervention using three-dimensional movies improves the usability of hands with distal radius fractures in daily life: a nonrandomized controlled trial in womenPLOS ONE

Dear Dr. Hamaguchi,

Thank you for submitting your manuscript to PLOS ONE. After careful consideration, we feel that it has merit but does not fully meet PLOS ONE’s publication criteria as it currently stands. Therefore, we invite you to submit a revised version of the manuscript that addresses the points raised during the review process.

We look forward to receiving your revised manuscript.

Kind regards,

Hanna Landenmark

Staff Editor

PLOS ONE

Journal Requirements:

   "This work was supported by JSPS KAKENHI Grant Number 26750226 and 22K21243."

Additional Editor Comments:

Please see the reports from three reviewers below. All reviewers have provided detailed comments for clarifying the manuscript, and both subject experts have raised concerns that many of the findings are largely expected. However, PLOS ONE considers manuscripts replicating previous works, if these studies are clearly labelled as such and relevant previous literature is discussed in the context of the findings. We invite you to address the reviewer concerns in a first revision, and ask that you work to assuage their doubts.

Reviewers' comments:

Reviewer's Responses to Questions

**Comments to the Author**

1. Is the manuscript technically sound, and do the data support the conclusions?

Reviewer #1: No

Reviewer #2: Partly

Reviewer #3: Yes

2. Has the statistical analysis been performed appropriately and rigorously? 

Reviewer #1: I Don't Know

Reviewer #2: No

Reviewer #3: I Don't Know

3. Have the authors made all data underlying the findings in their manuscript fully available?

Reviewer #1: Yes

Reviewer #2: Yes

Reviewer #3: Yes

4. Is the manuscript presented in an intelligible fashion and written in standard English?

Reviewer #1: Yes

Reviewer #2: No

Reviewer #3: No

5. Review Comments to the Author

Reviewer #1: Important note: This review pertains only to ‘statistical aspects’ of the study and so ‘clinical aspects’ [like medical importance, relevance of the study, ‘clinical significance and implication(s)’ of the whole study, etc.] are to be evaluated [should be assessed] separately/independently. Further please note that any ‘statistical review’ is generally done under the assumption that (such) study specific methodological [as well as execution] issues are perfectly taken care of by the investigator(s). This review is not an exception to that and so does not cover clinical aspects {however, seldom comments are made only if those issues are intimately / scientifically related & intermingle with ‘statistical aspects’ of the study}. Agreed that ‘statistical methods’ are used as just tools here, however, they are vital part of methodology [and so should be given due importance]. I look at the manuscript in/with statistical view point, other reviewer(s) look(s) at it with different angle so that in totality the review is very comprehensive. However, there should be efforts from authors side to improve (may be by taking clues from reviewer’s comments). Therefore, please do not limit the revision only (with respect) to comments made here.

COMMENTS: I noted that your ABSTRACT is well drafted (in my opinion), but is ‘assay type’. It is preferable [refer to item 1b of CONSORT checklist 2010: Structured summary of trial design, methods, results, and conclusions] to divide the ABSTRACT with small sections like ‘Objective(s)’, ‘Methods’, ‘Results’, ‘Conclusions’, etc. which is an accepted practice of most of the good/standard journals [including this one, though ‘The PLoS One Guidelines to Authors’ did not specify an Abstract format, it is desirable]. It will definitely be more informative then, I guess, whatever the article type may be [since the ‘Article Type’ is ‘Clinical Trial’ it is necessary indeed].

There are more issues (a few serious) about which I have different opinion. Such observations/concerns are given below:

This trial/study is on very important topic but title of the study [“Action observation intervention using three-dimensional movies improves the usability of hands with distal radius fractures in daily life”] is little confusing. It gives the impression that authors are interested [their main focus is on] in ‘three-dimensional video during ROM exercises’ whereas in lines 25-28 it is clearly mentioned that ‘This nonrandomized controlled trial tested the hypothesis that AOT improves hand-use difficulties during activities of daily living in patients with DRF’. Action observation intervention is the main intervention [lines 30-31: The primary outcome was the difficulty in using the fractured hand and line 27: AOT improves hand-use difficulties] which is evaluated but administered using three-dimensional movies. You may think of changing the title.

According to table-2 on page 158 of Jacob Cohen’s paper “A power primer” in Psychological Bulletin, 1992, vol.:112, pp 155-159 [which is a sort of summary of the excellent book by Cohen himself titled ‘Statistical power analysis for the behavioral sciences’, Academic Press, 1977, New York] even for medium effect size you need n=64 per group (type-I error=0.05, power=80%). This is given here in the context of account in lines 335-36 [where you said: “the final target sample size was set to be 34 cases in total, 17 cases in each group”]. Although you have quoted the reference number 28 { Schott N, Korbus H. Preventing functional loss during immobilization after osteoporotic wrist fractures in elderly patients : a randomized clinical trial. BMC Musculoskelet Disord. 2014; 15: 287 but this study is a randomized with three intervention groups & as I understand, it is only a protocol} please check the ‘control’ group to be used and all other issues. Was it (study described in reference 28) simillar [with respect to ‘design’] to yours?

Since (lines 340-41) “The primary outcome of this study was the PRWE score as a measure of the difficulty in using hands for ADLs” please check the level of measurement of data (to be) yielded by ‘The PRWE scale/questionnaire’. Though this measure/tool used is appropriate, is likely to yield data that are in [at the most] ‘ordinal’ level of measurement [and not in ratio level of measurement for sure {as the score two times higher does not indicate presence of that parameter/phenomenon as double (for example, a Visual Analogue Scales VAS score or say ‘depression’ score)}].

Then application of suitable non-parametric (or distribution free) test(s) is/are indicated/advisable [even if distribution may be ‘Gaussian’ (also called ‘normal’)]. Agreed that there is/are no non-parametric test(s)/technique(s) available to be used as alternative in all situation(s), but should be used whenever/wherever they are available. Therefore, in short use suitable non-parametric test(s)/technique(s) while dealing with data that are in ‘ordinal’ level of measurement even if [despite that] the distribution may be ‘Gaussian’.

This is pasted from one famous standard textbook on ‘Medical Research Methodology’ and authors may think of using ‘change scores’ by Friedman’s method.

In lines 367-69 [Missing values of measurements were assigned the mean value of the group and time period [39]]. Please check the correctness/context of quoting reference 39 {MillerJ. Another warning about median reaction time. J Exp Psychol Hum Percept Perform. 1988; 14(3): 539–543}. There seems to be some confusion there.

What is ‘df’ in table-1 [Table 1. Characteristics of the study participants]? As stated in lines 354-55 {the primary and secondary outcomes at baseline were compared using the Mann–Whitney U test} ‘U’ stands for Mann–Whitney test statistic which does not have ‘df’, then this df corresponds with which test? Basically, I did not understand ‘why do you used so much statistics [df; U; p values, Shapiro–Wilk p; Levene p] here in table-1. In this context, I request authors to read the following {again pasted from the same textbook on ‘Medical Research Methodology’}:

To provide a description of baseline characteristics is entirely reasonable (since it is clearly important in assessing to whom the results of the trial can be applied), however, statistical comparison of baseline characteristics when random allocation/assignment is used/done [often for good/standard/leading journals these days] is not required, because even if P-value(s) turn(s) out to be significant (while comparing baseline characteristics despite random allocation), it is, by definition, a false positive as you then are supposed to be testing ‘randomization’ then, which in any single trial may not balance all baseline characteristics (particularly when sample sizes are small). Remember that ‘randomization’ is a sort of ‘insurance’ and not a guarantee scheme. Authors may please refer to following articles:

References:

1. Stuart J. Pocock, et al., ‘Subgroup analysis, covariate adjustment and baseline comparisons in clinical trial reporting: current practice and problems’, Statistics in medicine, 2002; 21:2917–2930 [Particularly page 2927]

2. Harrington D, et al., ‘New guidelines for statistical reporting in the journal’, N Engl J Med 2019;381:285-6

[Important message (indirectly/ultimately indicated) from these articles: Never do any comparison with respect to ‘baseline’ characteristics {by applying statistical significance test(s)}, when allocation is done randomly].

However, Statistical comparison [only with respect to important/indicated variables] of baseline characteristics may be performed, to find out if analysis adjustment (say stratified analyses or else) is required with respect to these variables.

You should justify all statements (give purpose etc.), clarify details and properly interpret all the results (P-values). How you analysed data of table-2,3,4 {Table 2. Comparisons of gaps between conjectural and actual range of motions among groups at baseline, Table 3. Group comparisons for actual ROM, conjectural ROM, gap between actual and conjectural ROM, and PRWE score, Table 4. Comparisons of gaps between conjectural and actual range of motions between the study groups} is not clear. Note that while reporting Confidence Interval’ it is preferable to use ‘to’ [instead of ‘-‘ or ‘,’] between two numerical values, so as to avoid confusion with respect to negative (-) sign, for example, in table-3: for VF_Gap -13 (-28, -2) at Baseline, in AOT group, should be 95% CI: -28 to -2].

Because it is stated in lines 659 to 662 that “As this study was not a randomized controlled trial, the efficacy of this intervention must be verified in follow-up studies, including whether it is effective regardless of age and gender” and considering very small sample used for the study plus many other faults/limitations {example allocation – lines 217-222: The allocation was not randomized; 10 patients were assigned to the Non-AOT group from September 2013 to August 2014 and to the AOT group from January 2016 to October 2018 for adjustment of the experimental equipment (Ghost) depending on the time of the year}, best to call/classify this study as ‘PILOT’ and mention this in title.

Abstract of one article on guidelines for reporting non-randomised studies {Reeves BC, Gaus W. ‘Guidelines for reporting non-randomised studies’, Forsch Komplementarmed Klass Naturheilkd. 2004 Aug;11 Suppl 1:46-52. doi: 10.1159/000080576} reproduced below will hopefully give idea of ‘what is expected’:

Non-randomised studies (NRSs) are more susceptible to bias. The Consolidated Standards of Reporting Trials (CONSORT) statement was established to ensure that researchers report features of RCTs that must be considered when appraising their quality. CONSORT has improved the reporting of key information, highlighting missing key information for users. Researchers have a responsibility to report essential information that allows users to assess the susceptibility of NRS to selection, performance, detection and attrition bias. This paper considers criteria for reporting cohort studies: the rationale behind the CONSORT criteria for reporting of RCTs will be applied to cohort studies. Many of the criteria need no modification but application of others raise difficult issues for cohort studies, e.g.: description and standardisation of control and intervention treatments; description of the method of allocation; choice of prognostic factors to be collected; distinguishing between intended and provided treatments; collection of data on adverse and longterm outcomes; establishing a priori plans for analysis.

As you know (& is well-known) that while reporting [findings from randomised or non-randomised even] ‘Clinical Trial’ one should follow CONSORT guidelines. I request authors to kindly check/ensure that important items {like How sample size was determined (Item 7a), Random Sequence generation (Item 8a), Allocation concealment (Item 9), Blinding (Item 11a)} are included [since your article type is ‘Clinical Trial’, you are supposed to cover these items in the report]. Refer to lines 222-24: patients were assigned to the two groups alternately in the order of their prescriptions until the sample size was met is not a standard way of allocation (please note).

As pointed out in ‘important note’ above “This review pertains only to ‘statistical aspects’ of the study and so ‘clinical aspects’ should be assessed separately/independently [one should carefully consider/look at the clinical implications of the study]. In my opinion, to rescue this article (which seems quite difficult, if not impossible), large amount of re-vision (re-drafting) may be needed. However, please do not limit the revision only (with respect) to comments made here. More improvement is expected. ‘Major revision’ is recommended [in lieu of plain rejection, assuming that the respected editor would like to give chance to authors for improvement of the manuscript].

Reviewer #2: The authors report a non-randomized study assessing the effect of virtual rehabilitation vs standard care after DRF. The authors perform a lot of measurements and find some faster recovery before 12 weeks after surgery.

A lot of data was collected. The main problem is, in my practice I don't send people generally to hand therapy after DRF. Several RCTs have shown that there is no difference, or people do better without. So I'm not sure what this virtual rehab truly adds to clinical practice.

Introduction

First 2 paragraphs can be omitted, or shortened into a few sentences.

Line 75-78: seems more like the authors opinion than fact. People generally find their own path of recovery, and some recover faster than others. I don’t think you can set goals like that. Some people are back to housework much faster than 7-8 weeks. You don’t want to slow them down with these goals.

Several randomized studies show no difference between people ‘rehabilitating’ with a hand therapist vs on their own. Some studies show people who do it on their own recover faster. Please include this in your introduction and discussion.

People can start some weightbearing activities right after volar plate fixation. I have people start finger motion right away, and wrist motion after 2 weeks. There are great early active rehab protocols out there.

Internal and external rotation, you mean pronation and supination?

You state: “Action observation therapy (AOT) is an effective means of preventing DDP in patients”, that remains to be tested no?

I think the introduction can be shortened.

I think we need more information or a video on the exact Ghost program.

Results

You exclude a seemingly large amount of data based on outliers. Can you mention a sensitivity analysis with those outliers? Are results the same?

Consider grouping some outcomes. If you take every ROM separately, you are performing a lot of analysis. Why not use total ROM for example as a single outcome?

Discussion

Is all of this worth a slight improvement at 4 weeks, while at 3 months there isn’t much of a difference?

This is not a RCT. Keep in mind that previous RCTs show no benefit to hand therapy after DRF in most people.

Generally the bio-psycho-social model of health is preferred. I would discuss the role of mental health and ROM/PRWE after distal radius fracture. Some studies show it explains 70% of the variation in recovery. Your focus on motor cortex brain blood flow is too limited, it’s only the bio part of health, without the psycho-social which seems to be more important.

“It has been reported that the flexor digitorum longus tendon, which is elongated during dorsiflexion, is impaired after volar locking plate surgery”. I don’t know which tendon this is, and I am also not sure the muscle is impaired. Possibly FPL, but that should limit ROM too much.

“square iliacus muscle”, do you mean pronator quadratus?

It’s a bit unclear to me how the virtual model truly helps increase motion.

Reviewer #3: This study aims to compare two different types of postoperative physical therapy after surgery for distal radius fractures. The study spanned eight years and included 35 patients, all of whom were female and had fractures in the right hand, their dominant hand. This indicates that the study rigorously controlled for various factors that could influence surgical outcomes. The study showed patients who underwent AOT had better PRWE scores in weeks 4 and 8. Such results are not very surprising, mainly because the comparison depends on the control group (non-AOT) and the postoperative rehabilitation protocol (unfortunately, not described in this study). Many past studies have already told us that early mobilization helps with short-term outcomes, but there won't be much difference in intermediate-term results, which this study confirms.

Introduction:

1. Please shorten the introductory paragraph to under 500 words, providing sufficient background knowledge and detailing the clinical dilemma faced by the surgeon.

2. Lines 90-100 and 118-119: Please replace Internal Rotation/External Rotation with pronation/supination.

3. Line 105: It is quite strange that Figure 2 is placed here.

4. Line 178: Where is Figure 3?

5. Lines 176-178, please pose a clear question this manuscript seeks to answer: Can AOT in the early postoperative period of a distal radius fracture (DRF) promote quicker functional recovery? What are the primary and secondary outcomes for measuring its effectiveness?

Methods:

1. This study was not randomized, and the number of participants was small. Please describe the fracture patterns of the distal radius fractures included in the study, whether only volar locking plates were used, or whether any patients underwent additional plating for specific fragment fixation.

2. Compared to AOT, does the non-AOT group follow a specific standardized protocol?

3. Was bony consolidation confirmed by X-rays after surgery? Was there any difference between the two groups?

Results:

1. Figure legends scattered irregularly throughout the manuscript

2. The presentation of tables is very chaotic, difficult to read the figures.

Discussion:

1. Line 582: What is the square iliacus muscle?

2. Please discuss why there is no difference between the two groups in 12 weeks.

6. PLOS authors have the option to publish the peer review history of their article (what does this mean?). If published, this will include your full peer review and any attached files.

Reviewer #1: No

Reviewer #2: **Yes: **Teun Teunis

Reviewer #3: **Yes: **Wen-Chih Liu

---

## [Author Response · Author response to Decision Letter 0]

8 May 2024

May 4th, 2024

Hanna Landmark

Editor

PLOS ONE

Dear Dr. Hanna Landmark,

We would like to thank you and the reviewers for your careful review of our manuscript. The insightful statistical perspective of the reviewers offered invaluable suggestions that greatly enriched our work. As the first author, Kengo Usuki, is a novice researcher, your feedback has been particularly valuable in enhancing our understanding and approach. This study has proven to be a beneficial experience for us, thanks in large part to your contributions.

The manuscript has been rechecked and the necessary changes have been made in accordance with the reviewers’ suggestions. The responses to all comments have been prepared and given below. For your convenience, all changes in the revised manuscript are highlighted in yellow. 

Thank you for your consideration. I look forward to hearing from you.

Sincerely,

Toyohiro Hamaguchi, Ph.D

Department of Rehabilitation,

Graduate School of Health Sciences,

Saitama Prefectural University,

820, Sannomiya, Koshigaya, Saitama, Japan

Phone: +81-48-971-4125

Fax: +81-48-973-4807

Email: hamaguchi-toyohiro@spu.ac.jp

Response for reviewer 1

Reviewer 1, Comment＃1 

I noted that your ABSTRACT is well drafted (in my opinion), but is ‘assay type’. It is preferable [refer to item 1b of CONSORT checklist 2010: Structured summary of trial design, methods, results, and conclusions] to divide the ABSTRACT with small sections like ‘Objective(s)’, ‘Methods’, ‘Results’, ‘Conclusions’, etc. which is an accepted practice of most of the good/standard journals [including this one, though ‘The PLoS One Guidelines to Authors’ did not specify an Abstract format, it is desirable].

Response to Comment #1 

As per your comment, we have added the “Objective,” “Trial design,” “Method,” “Results,” and “Conclusions” headings to the abstract in accordance with the CONSORT 2010 guidelines.

Reviewer 1, Comment＃2 

This trial/study is on very important topic but title of the study [“Action observation intervention using three-dimensional movies improves the usability of hands with distal radius fractures in daily life”] is little confusing. It gives the impression that authors are interested [their main focus is on] in ‘three-dimensional video during ROM exercises’ whereas in lines 25-28 it is clearly mentioned that ‘This nonrandomized controlled trial tested the hypothesis that AOT improves hand-use difficulties during activities of daily living in patients with DRF’. Action observation intervention is the main intervention [lines 30-31: The primary outcome was the difficulty in using the fractured hand and line 27: AOT improves hand-use difficulties] which is evaluated but administered using three-dimensional movies. You may think of changing the title.

Response to Comment #2 

As per your suggestion, we have revised the title to “Pilot study: Action observation intervention using three-dimensional movies improves the usability of hands with distal radius fractures in daily life: a nonrandomized controlled trial in women.”

Reviewer 1, Comment＃3 

According to table-2 on page 158 of Jacob Cohen’s paper “A power primer” in Psychological Bulletin, 1992, vol.:112, pp 155-159 [which is a sort of summary of the excellent book by Cohen himself titled ‘Statistical power analysis for the behavioral sciences’, Academic Press, 1977, New York] even for medium effect size you need n=64 per group (type-I error=0.05, power=80%). This is given here in the context of account in lines 335-36 [where you said: “the final target sample size was set to be 34 cases in total, 17 cases in each group”]. Although you have quoted the reference number 28 { Schott N, Korbus H. Preventing functional loss during immobilization after osteoporotic wrist fractures in elderly patients : a randomized clinical trial. BMC Musculoskelet Disord. 2014; 15: 287 but this study is a randomized with three intervention groups & as I understand, it is only a protocol} please check the ‘control’ group to be used and all other issues. Was it (study described in reference 28) simillar [with respect to ‘design’] to yours? 

Response to Comment #3 

We agree that our sample size was small and the Patient Rated Wrist Evaluation (PRWE) scores did not have a normal distribution. Initially, we had considered using analysis of variance. Instead, PRWE, as reported in a previous study (Nadia Schott. 2014), used as the outcome. We believe that the designs are similar. In the future, we plan to continue collecting samples and further validate this finding by using variance analysis.

Reference:

Schott N, Korbus H. Preventing functional loss during immobilization after osteoporotic wrist fractures in elderly patients: A randomized clinical trial. BMC Musculoskelet Disord. 2014;15: 287.

Reviewer 1, Comment＃4 

Since (lines 340-41) “The primary outcome of this study was the PRWE score as a measure of the difficulty in using hands for ADLs” please check the level of measurement of data (to be) yielded by ‘The PRWE scale/questionnaire’. Though this measure/tool used is appropriate, is likely to yield data that are in [at the most] ‘ordinal’ level of measurement [and not in ratio level of measurement for sure {as the score two times higher does not indicate presence of that parameter/phenomenon as double (for example, a Visual Analogue Scales VAS score or say ‘depression’ score)}]. Then application of suitable non-parametric (or distribution free) test(s) is/are indicated/advisable [even if distribution may be ‘Gaussian’ (also called ‘normal’)]. Agreed that there is/are no non-parametric test(s)/technique(s) available to be used as alternative in all situation (s), but should be used whenever/wherever they are available. Therefore, in short use suitable non-parametric test(s)/technique(s) while dealing with data that are in ‘ordinal’ level of measurement even if [despite that] the distribution may be ‘Gaussian’. 

Response to Comment #4 

Thank you for your important point. We first conducted the Friedman test on the VR group. Similar to the generalized linear model (GLM) results, only PRWE showed a cautious difference, depending on the period (Table).

The results of the Friedman test are published in a pictorial file in a separately submitted reply file.

Post hoc test: the Durbin–Conover test. 

PRWE_Total: A significant difference exists for baseline to 1 month after surgery (p<0.01), 1 month after surgery to 2 months after surgery (p<0.01), and 2 months after surgery to 3 months after surgery (p<0.01). 

PRWE_Pain: A significant difference exists for baseline to 1 month after surgery (p=0.01), 1 month after surgery to 2 months after surgery (p=0.01), and 2 months after surgery to 3 months after surgery (p<0.01). 

PRWE_Specific: A significant difference exists for baseline to 1 month after surgery (p<0.01), 1 month after surgery to 2 months after surgery (p<0.01), and 2 months after surgery to 3 months after surgery (p<0.01). 

PRWE_Usual: A significant difference exists for baseline to 1 month after surgery (p<0.01) and 1 month after surgery to 2 months after surgery (p<0.01).

Actual_ROM_Volar_Flexion: A significant difference exists for baseline to 1 month after surgery (p<0.01) and 1 month after surgery to 2 months after surgery (p<0.01). 

Actual_ROM_Dorsal_Flexion: A significant difference exists for baseline to 1 month after surgery (p<0.01), 1 month after surgery to 2 months after surgery (p<0.01), and 2 months after surgery to 3 months after surgery (p=0.04). 

Actual_ROM_Pronation: A significant difference exists for baseline to 1 month after surgery (p<0.01) and 1 month after surgery to 2 months after surgery (p<0.01). 

Actual_ROM_Supination: A significant difference exists for baseline to 1 month after surgery (p<0.01).

Among all terms in the post hoc test, no significant difference existed for Gap_Volar_Flexion, Gap_Dorsal_Flexion, Gap_pronation, and Gap Supination. 

However, the Friedman test cannot be used to compare the VR group and the Control group. The results of GLM are similar to those of PRWE when comparing between groups, and we believe that this is close to the Friedman's test. GLM can specify the distribution of the response variables through link functions and probability distributions. This allows them to be applicable to various types of data that are not limited to normal distribution including binomial distribution, Poisson distribution, and negative binomial distribution. This means that the model can be customized according to the specific data type or research needs. Additionally, research papers have analyzed the values of the PRWE using ANOVA. (1, 2) Therefore, we have analyzed the data of this study by adopting GLM.

References:

1. Watson N, Haines T, Tran P, Keating JL. A comparison of the effect of one, three, or six weeks of immobilization on function and pain after open reduction and internal fixation of distal radial fractures in adults a randomized controlled trial. Journal of Bone and Joint Surgery - American Volume. 2018;100(13):1118–25. 

2. Thorninger R, Romme KL, Wæver D, Henriksen MB, Tjørnild M, Lind M, et al. Posttraumatic arthritis and functional outcomes of nonoperatively treated distal radius fractures after 3 years. Sci Rep. 2023 Dec 1;13(1). 

Reviewer 1, Comment＃5 

In lines 367-69 [Missing values of measurements were assigned the mean value of the group and time period [39]]. Please check the correctness/context of quoting reference 39 {MillerJ. Another warning about median reaction time. J Exp Psychol Hum Percept Perform. 1988; 14(3): 539–543}. There seems to be some confusion there. 

Response to Comment #5 

Thank you for pointing this out. The text has been corrected. Please refer to line 223.

Reviewer 1, Comment＃6 

What is ‘df’ in table-1 [Table 1. Characteristics of the study participants]? As stated in lines 354-55 {the primary and secondary outcomes at baseline were compared using the Mann–Whitney U test} ‘U’ stands for Mann–Whitney test statistic which does not have ‘df’, then this df corresponds with which test? Basically, I did not understand ‘why do you used so much statistics [df; U; p values, Shapiro–Wilk p; Levene p] here in table-1. In this context, I request authors to read the following {again pasted from the same textbook on ‘Medical Research Methodology’}: To provide a description of baseline characteristics is entirely reasonable (since it is clearly important in assessing to whom the results of the trial can be applied), however, statistical comparison of baseline characteristics when random allocation/assignment is used/done [often for good/standard/leading journals these days] is not required, because even if P-value(s) turn(s) out to be significant (while comparing baseline characteristics despite random allocation), it is, by definition, a false positive as you then are supposed to be testing ‘randomization’ then, which in any single trial may not balance all baseline characteristics (particularly when sample sizes are small). Remember that ‘randomization’ is a sort of ‘insurance’ and not a guarantee scheme. Authors may please refer to following articles:

Response to Comment #6

We acknowledge this comment. We have revised Table 1. With regard to the use of df, we initially believed its use was necessary, as with the t-test. However, upon reflection and considering the nature of our study as a nonrandomized controlled trial, we have concluded that it would be more appropriate to delineate normality and homoscedasticity as the primary rationale for our chosen analysis method. Therefore, df has been deleted and the Mann–Whitney U information has been retained. Details regarding normality and homoscedasticity remain within the main text for clarity and transparency. 

Reviewer 1, Comment #7

You should justify all statements (give purpose etc.), clarify details and properly interpret all the results (P-values). How you analysed data of table-2,3,4 {Table 2. Comparisons of gaps between conjectural and actual range of motions among groups at baseline, Table 3. Group comparisons for actual ROM, conjectural ROM, gap between actual and conjectural ROM, and PRWE score, Table 4. Comparisons of gaps between conjectural and actual range of motions between the study groups} is not clear. Note that while reporting Confidence Interval’ it is preferable to use ‘to’ [instead of ‘-‘ or ‘,’] between two numerical values, so as to avoid confusion with respect to negative (-) sign, for example, in table-3: for VF_Gap -13 (-28, -2) at Baseline, in AOT group, should be 95% CI: -28 to -2].

Response to Comment #7

The purpose has been added to the explanations in Tables 2–4 in the main text (Table 2; Table 3; Table 4). The interpretation of the results has also been described. In Table 2 and Table 4, “Gaps between conjectural and actual range of motions” was initially written as “Difference between.” This text has been changed to “Gap between” In Table 3, the interpretation of the results has been provided.

Reviewer 1, Comment＃8

Because it is stated in lines 659 to 662 that “As this study was not a randomized controlled trial, the efficacy of this intervention must be verified in follow-up studies, including whether it is effective regardless of age and gender” and considering very small sample used for the study plus many other faults/limitations {example allocation – lines 217-222: The allocation was not randomized; 10 patients were assigned to the Non-AOT group from September 2013 to August 2014 and to the AOT group from January 2016 to October 2018 for adjustment of the experimental equipment (Ghost) depending on the time of the year}, best to call/classify this study as ‘PILOT’ and mention this in title.

Response to Comment #8

We have added the term “Pilot study” to the title.

Reviewer 1, Comment＃9 & 10

Abstract of one article on guidelines for reporting non-randomised studies {Reeves BC, Gaus W. ‘Guidelines for reporting non-randomised studies’, Forsch Komplementarmed Klass Naturheilkd. 2004 Aug;11 Suppl 1:46-52. doi: 10.1159/000080576} reproduced below will hopefully give idea of ‘what is expected’: Non-randomised studies (NRSs) are more susceptible to bias. The Consolidated Standards of Reporting Trials (CONSORT) statement was established to ensure that researchers report features of RCTs that must be considered when appraising their quality. CONSORT has improved the reporting of key information, highlighting missing key information for users. Researchers have a responsibility to report essential information that allows users to assess the susceptibility of NRS to selection, performance, detection and attrition bias. This paper considers criteria for reporting cohort studies: the rationale behind the CONSORT criteria for reporting of RCTs will be applied to cohort studies. Many of the criteria need no modification but application of others raise difficult issues for cohort studies, e.g.: description and standardisation of control and intervention treatments; description of the method of allocation; choice of prognostic factors to be collected; distinguishing between intended and provided treatments; collection of data on adverse and longterm outcomes; establishing a priori plans for analysis.

As you know (& is well-known) that while reporting [findings from randomised or non-randomised even] ‘Clinical Trial’ one should follow CONSORT guidelines. I request authors to kindly check/ensure that important items {like How sample size was determined (Item 7a), Random Sequence generation (Item 8a), Allocation concealment (Item 9), Blinding (Item 11a)} are included [since your article type is ‘Clinical Trial’, you are supposed to cover these items in the report]. Refer to lines 222-24: patients were assigned to the two groups alternately in the order of their prescriptions until the sample size was met is not a standard way of allocation (please note). As pointed out in ‘important note’ above “This review pertains only to ‘statistical aspects’ of the study and so ‘clinical aspects’ should be assessed separately/independently [one should carefully consider/look at the clinical implications of the study]. In my opinion, to rescue this article (which seems quite difficult, if not impossible), la

---

## [Decision Letter · Decision Letter 1]

20 May 2024

PONE-D-23-33804R1Pilot study: Action observation intervention using three-dimensional movies improves the usability of hands with distal radius fractures in daily life: A nonrandomized controlled trial in womenPLOS ONE

Dear Dr. Hamaguchi,

Thank you for submitting your manuscript to PLOS ONE. After careful consideration, we feel that it has merit but does not fully meet PLOS ONE’s publication criteria as it currently stands. Therefore, we invite you to submit a revised version of the manuscript that addresses the points raised during the review process.

We look forward to receiving your revised manuscript.

Kind regards,

Jianxun Ding, Ph.D.

Academic Editor

PLOS ONE

Journal Requirements:

Reviewers' comments:

Reviewer's Responses to Questions

**Comments to the Author**

1. If the authors have adequately addressed your comments raised in a previous round of review and you feel that this manuscript is now acceptable for publication, you may indicate that here to bypass the “Comments to the Author” section, enter your conflict of interest statement in the “Confidential to Editor” section, and submit your "Accept" recommendation.

Reviewer #1: All comments have been addressed

2. Is the manuscript technically sound, and do the data support the conclusions?

Reviewer #1: (No Response)

3. Has the statistical analysis been performed appropriately and rigorously? 

Reviewer #1: (No Response)

4. Have the authors made all data underlying the findings in their manuscript fully available?

Reviewer #1: (No Response)

5. Is the manuscript presented in an intelligible fashion and written in standard English?

Reviewer #1: (No Response)

6. Review Comments to the Author

Reviewer #1: COMMENTS: Since most of the comments made on earlier draft are considered positively & attended, I recommend the acceptance after very minor revision. The manuscript now has achieved acceptable level, in my opinion (just need to mention few things/limitations taking clue from following). As the study is now labelled ‘pilot’ in nature, sample size [and/or rigorous methodology] is/are not a big issue.

However, it is very important to note that ‘RANDOMIZATION’ described in lines 117-123 [Experimental procedures: Study participants were assigned to one of two groups: the group that received AOT and the group that received the usual rehabilitation without AOT (Non-AOT). The allocation was not randomized: 10 patients were assigned to the Non-AOT group from September 2013 to August 2014 and to the AOT group from January 2016 to October 2018 to adjust the experimental equipment (i.e., Ghost), depending on the time of the year. The patients were thereafter assigned to the two groups alternately in the order of their prescriptions until the sample size was met in March 2021.] is not scientific and so not agreeable. But since the study is now labelled ‘pilot’ in nature, randomization described is not that important/vital and so can be ignored/accepted.

Also note that the responses to all the comments (by all reviewers) are/were with unnecessary details and/or references {on many occasions}. They are not ‘precise’ as desired. At times very lengthy.

7. PLOS authors have the option to publish the peer review history of their article (what does this mean?). If published, this will include your full peer review and any attached files.

Reviewer #1: No

---

## [Author Response · Author response to Decision Letter 1]

31 May 2024

Response for reviewer 1

Reviewer #1: COMMENTS: Since most of the comments made on earlier draft are considered positively & attended, I recommend the acceptance after very minor revision. The manuscript now has achieved acceptable level, in my opinion (just need to mention few things/limitations taking clue from following). As the study is now labelled ‘pilot’ in nature, sample size [and/or rigorous methodology] is/are not a big issue.

However, it is very important to note that ‘RANDOMIZATION’ described in lines 117-123 [Experimental procedures: Study participants were assigned to one of two groups: the group that received AOT and the group that received the usual rehabilitation without AOT (Non-AOT). The allocation was not randomized: 10 patients were assigned to the Non-AOT group from September 2013 to August 2014 and to the AOT group from January 2016 to October 2018 to adjust the experimental equipment (i.e., Ghost), depending on the time of the year. The patients were thereafter assigned to the two groups alternately in the order of their prescriptions until the sample size was met in March 2021.] is not scientific and so not agreeable. But since the study is now labelled ‘pilot’ in nature, randomization described is not that important/vital and so can be ignored/accepted.

Also note that the responses to all the comments (by all reviewers) are/were with unnecessary details and/or references {on many occasions}. They are not ‘precise’ as desired. At times very lengthy.

Response to Comment #1 

Thank you for your insightful comments and for acknowledging the improvements made in the manuscript. We appreciate your recommendation for acceptance after minor revisions and agree that this pilot study's nature makes the sample size and rigorous methodology less critical. 

We understand that true randomization is the gold standard for reducing bias and increasing the robustness of study results. However, given the nature of the pilot for this study, we believe that this approach was the only way to achieve the exploratory objectives of this initial survey. We acknowledge this as a limitation and recognize that true randomization would enhance the robustness of the findings. Nevertheless, the pilot nature of this study aims to provide preliminary insights that could inform the design of future randomized controlled trials. 

Additionally, we take feedback on our responses to reviewers' comments very seriously. We have reviewed and revised our manuscript to ensure that our responses were concise and accurate, focusing on directly addressing the reviewers' concerns without including unnecessary details or references. 

Thank you once again for your valuable feedback, which has significantly contributed to the improvement of our manuscript.

---

## [Decision Letter · Decision Letter 2]

17 Jun 2024

Pilot study: Action observation intervention using three-dimensional movies improves the usability of hands with distal radius fractures in daily life: A nonrandomized controlled trial in women

PONE-D-23-33804R2

Dear Dr. Hamaguchi,

We’re pleased to inform you that your manuscript has been judged scientifically suitable for publication and will be formally accepted for publication once it meets all outstanding technical requirements.

Kind regards,

Jianxun Ding, Ph.D.

Academic Editor

PLOS ONE

Additional Editor Comments (optional):

The revised manuscript is ready for publication.

Reviewers' comments:

Reviewer's Responses to Questions

**Comments to the Author**

1. If the authors have adequately addressed your comments raised in a previous round of review and you feel that this manuscript is now acceptable for publication, you may indicate that here to bypass the “Comments to the Author” section, enter your conflict of interest statement in the “Confidential to Editor” section, and submit your "Accept" recommendation.

Reviewer #1: All comments have been addressed

2. Is the manuscript technically sound, and do the data support the conclusions?

Reviewer #1: (No Response)

3. Has the statistical analysis been performed appropriately and rigorously? 

Reviewer #1: (No Response)

4. Have the authors made all data underlying the findings in their manuscript fully available?

Reviewer #1: (No Response)

5. Is the manuscript presented in an intelligible fashion and written in standard English?

Reviewer #1: (No Response)

6. Review Comments to the Author

Reviewer #1: COMMENTS: Since all of the comments made on earlier draft are considered positively & attended, I recommend the acceptance. The manuscript now has achieved acceptable level, in my opinion. In fact, I had already (based on original draft) passed recommendation for acceptance of this manuscript after minor revision as the pilot study's nature makes the sample size and rigorous methodology less critical.

Only note that in the title [Pilot study: Action observation intervention using three-dimensional movies improves the usability of hands with distal radius fractures in daily life: A nonrandomized controlled trial in women] there are two colons [‘:’] used {which may not acceptable to grammar experts}. Please consider modification.

7. PLOS authors have the option to publish the peer review history of their article (what does this mean?). If published, this will include your full peer review and any attached files.

Reviewer #1: No

---

## [Editor Report · Acceptance letter]

1 Jul 2024

PONE-D-23-33804R2 

PLOS ONE

Dear Dr. Hamaguchi, 

I'm pleased to inform you that your manuscript has been deemed suitable for publication in PLOS ONE. Congratulations! Your manuscript is now being handed over to our production team.

Kind regards, 

on behalf of

Dr. Jianxun Ding 

Academic Editor

PLOS ONE